# Development of the Choices 5-Level Criteria to Support Multiple Food System Actions

**DOI:** 10.3390/nu13124509

**Published:** 2021-12-16

**Authors:** Gianluca Tognon, Belen Beltramo, Rutger Schilpzand, Lauren Lissner, Annet J. C. Roodenburg, Rokiah Don, Krishnapillai Madhavan Nair, Ngozi Nnam, Bruce Hamaker, Herbert Smorenburg

**Affiliations:** 1Choices International Foundation, 2501 HE The Hague, The Netherlands; gianluca.tognon@choicesprogramme.org (G.T.); Belen.Beltramo@choicesprogramme.org (B.B.); rutger.schilpzand@choicesprogramme.org (R.S.); 2School of Health Sciences, University of Skövde, 541 28 Skövde, Sweden; 3Institute of Medicine, School of Public Health and Community Medicine, Sahlgrenska Academy, University of Gothenburg, 405 30 Gothenburg, Sweden; lauren.lissner@gu.se; 4Department of Nutrition and Health HAS, University of Applied Sciences, 5200 MA ’s-Hertogenbosch, The Netherlands; A.Roodenburg@has.nl; 5Nutrition and Dietetics Division, International Medical University, Kuala Lumpur 57000, Malaysia; rokiah.don@gmail.com; 6Former Scientist, National Institute of Nutrition (ICMR), Choices International Scientific Committee, Hyderabad 500007, India; nairthayil@gmail.com; 7Department of Nutrition & Dietetics, University of Nigeria, Nsukka 410001, Nigeria; ngnnam@yahoo.com; 8Department of Food Science, Purdue University, West Lafayette, IN 47907, USA; hamakerb@purdue.edu

**Keywords:** criteria for healthy food products, front-of-pack labelling FOPL, nutrient profiling, nutrition policy, non-communicable diseases, malnutrition, indicator foods

## Abstract

In 2008, the Choices International Foundation developed its logo criteria, identifying best-in-class food products. More advanced, global and graded nutrient profiling systems (NPSs) are needed to substantiate different national nutrition policies. The objective of this work was to extend Choices NPS to identify five levels of the healthiness of food products, so that the Choices NPS can also be used to support other nutrition policies, next to front-of-pack labelling. Based on the same principles as the previous logo criteria, four sets of threshold criteria were determined using a combination of compliance levels, calculated from a large international food group-specific database, the Choices logo criteria, and WHO-NPSs developed to restrict marketing to children. Validation consisted of a comparison with indicator foods from food-based dietary guidelines from various countries. Some thresholds were adjusted after the validation, e.g., because intermediate thresholds were too lenient. This resulted in a new international NPS that can be applied to different contexts and to support a variety of health policies, to prevent both undernutrition and obesity. It can efficiently evaluate mixed food products and represents a flexible tool, applicable in various settings and populations.

## 1. Introduction

Malnutrition is a hallmark of both obesity and undernutrition, and it is part of a vicious cycle, involving poverty and disease [1]. Despite many risk factors for all forms of malnutrition being known, no government or institution has implemented existing evidence-based policy recommendations in such a way that the malnutrition epidemic figures have been reversed. Swinburn et al. [2] defined this situation as a form of ‘policy inertia’, attributable to the collective effects of lack of political leadership in the field of nutrition and public health, intense lobbying by interest groups and businesses, and a lack of demand for action by the general public. Efforts to address malnutrition by working in collaboration with industry through self-regulation appears insufficiently effective [2]. Quasi-regulatory approaches, such as front-of-pack labelling (FOPL) initiatives, where governments are in the lead and participation of the industry is on voluntary basis, can positively influence food choice [3,4]. However, their full implementation may often take a long time. Nutrition interventions by public health institutions are often focused on individual aspects of malnutrition and implemented with varying governance and funding.

A shift towards interventions and policies tackling both obesity and undernutrition (by the so-called ‘double-duty actions’), coordinated by governments and based on independent science-based goals and tools, is needed to improve global health, while optimizing the resources needed to achieve this result. Hawkes et al. [5] defined ten priority double-duty actions designed to reverse the burden of undernutrition, obesity, and diet-related non-communicable diseases, in a holistic way (Figure 1).

Many double-duty actions would benefit from a nutrient profiling system (NPS); an objective method for evaluating the nutritional quality of different food products, which should be suitable for multiple applications. These applications include restrictions of the marketing of unhealthy foods to children, claim regulations, guidance for FOPL systems, product reformulation, and food quality standards for school and health facility canteens [6]. By rewarding stepwise improvements in food composition, a valid NPS would ultimately stimulate food product reformulation. For infants a dedicated NPS could be developed, considering their specific nutritional needs from complementary foods. However, the application of many NPSs is still highly fragmented and usually oriented to single actions, rather than to curbing malnutrition in general. Choices believes that coherent and effective governmental policies for the promotion of healthier diets should be guided based on a global, science-based NPS that classifies food products according to specific nutrient contents and is able to support multiple interventions, to promote the consumption of healthier, and discourage the consumption of less healthy, foods. Such a global NPS should be inspired by the same principles (e.g., the encouragement of a lower content of less healthy nutrients, such as SFA) but, at the same time, be customizable to local or regional contexts, according to the food culture specificities, local nutritional issues, national food standards, and the foods available in the local market.

The objective of the present work was, therefore, to propose a new science-based NPS to classify food products based on their nutrient contents, which is applicable worldwide and able to support multiple health interventions (e.g., double-duty actions). Choices has chosen a five-level practical and actionable system that can readily be applied to define which foods should be encouraged and which should be discouraged through various policies. Such a five-level NPS can support positive FOPL, as the Choices criteria were originally developed for, but graded five- or three-level systems of FOPL can also be substantiated. Moreover, other food system actions, such as restricting advertising to children, reformulation, financial incentives and disincentives, school food environments, fortification, and claims can be supported, preferably in combination with a mandatory graded FOPL system (see Figure 2). By differentiating between basic and non-basic food groups in how the Choices levels are used to support food system actions, these actions are consistent with and complementary to food-based dietary guidelines.

Previous versions of the Choices NPS and related programs have been described in the literature [7,8]. The Choices NPS was developed to identify food products eligible for a positive FOPL, to encourage consumers to select healthier food options and to encourage food companies to reformulate their food products. Various studies have shown the efficacy of Choices programs [9,10,11]. However, as shown in Figure 2, we believe that the Choices NPS should, not only support FOPL, but also provide guidance to other double-duty actions.

Recently, the World Health Organization (WHO) published a report to inform policy-makers on the development and implementation of interpretive FOPL policies across the WHO European region [12]. In 2020, they developed a manual on FOPL, in which the WHO advocates using a government-approved FOPL system, with both positive and negative evaluative judgments and more stringent implementation and monitoring programs [13]. Therefore, in response to the WHO’s suggestions and with the ambition to support a series of double-duty actions, and not only a FOPL system, Choices decided to extend its criteria to classify food products into five levels of healthiness, instead of two.

## 2. Materials and Methods

### 2.1. Development of Intermediate Five-Level Criteria

The five-level criteria framework is based on the same principles as the previous version of the Choices logo criteria [8]. The same food group definitions, the same concept of basic and non-basic food groups, the calculation of compliance levels in the market to define the criteria, the same key nutrients, and the same insignificancy levels are used. Similarly to the previous Choices single-level criteria, to be classified in a certain healthiness level, a product needs to comply with all corresponding individual nutrient criteria. For those food groups for which an energy criterion is specified, energy density is expressed in kcal/100 g, in addition to the previous energy per portion size criteria. Whilst the logo criteria defined one set of nutrient threshold levels (or cut-points), to distinguish two healthiness levels, the new criteria consist of four thresholds (T1, T2, T3, and T4), to distinguish five levels of healthiness; with level 1 the healthiest products, and level 5 the least healthy products. To define these thresholds, a few aspects were considered. Similarly to the previous ones, the new criteria must be realistic, and this is the reason why we tested compliance on a database of food products available in markets.

The previously described [8] large international product-specific food composition databases of the George Institute and others were further cleaned up, resulting in a database of over 64,700 food products from eight countries. This database was initially used to determine all nutrient thresholds T1–T4, so that the corresponding compliance levels C1–C4 were approximately 20%, 40%, 60%, and 80%, respectively. In order to ensure that compliance was not determined by one nutrient, but that all nutrient thresholds would be equally restrictive, nutrient thresholds were calculated such that compliance levels against a single nutrient threshold would not differ by more than 5% from each other in each food group.

The new criteria should, as much as possible, align with the Choices logo criteria, as well as other existing international standards. Therefore, as a starting point, the logo criteria were used as benchmark for T1. T3 was considered as the threshold discerning between products that should not, or may, be restricted. Therefore, the five different regional WHO NPSs to restrict marketing to children [14,15,16,17,18] were considered as input for T3. The WHO NPSs have been recently developed to support a particular restrictive measure and are based on a food group-specific NPS. The five WHO NPSs provide thresholds for critical nutrients per food group in each WHO region. These nutrients are total fat, saturated fat, total sugars, added sugars, and/or sodium. The WHO European NPS [14] was the initial model, from which the other four models were derived. The WHO East Mediterranean region and WHO West Pacific Region models are very similar to the European model; however, the African region [18] and Southeast Asian region [16] models deviate in some food group definitions and threshold levels. The key characteristics of these NPSs are summarized in Appendix A and were used as the reference models to define T3 for basic food groups.

The process of establishing each threshold is illustrated in Figure 3, and an example of how this worked in practice for the food group ‘processed beans and legumes’ is given in Figure 4.

When the food groups defined by the WHO and Choices were not aligned with each other, WHO was not used. Otherwise, the thresholds from the three WHO NPSs were compared with Choices logo criteria. This comparison led to four possible situations: two in which the WHO were used and two in which the WHO were not used. Whether these thresholds were realistic was decided by looking at compliance levels C1 and C3: (1) one or more nutrient thresholds of the WHO NPSs were at least 50% less restrictive than the comparable threshold of the Choices logo criteria. In that case, the WHO thresholds were used as input for T3, and the logo criteria were used for T1; (2) when none of the nutrient thresholds in the WHO NPSs were at least 50% less restrictive than the logo criteria, it was considered realistic to make the T1 thresholds 50% stricter than the WHO nutrient threshold(s), and one or more nutrient thresholds of the WHO NPSs were used for T3; (3) it was considered unrealistic to make T1 at least 50% stricter than WHO NPSs; hence, the logo criteria were used for T1, the WHO-NPS was ignored, and T3 was instead calculated by interpolation; (4) for the same reason as mentioned in (3), the WHO NPSs were not used, but T1 was made stricter than the Choices logo criteria, based on realistic compliance levels. Note that all these decisions led to at least a determination of T1, and in some cases also a (partial) determination of T3.

If the WHO NPs had been used for some nutrient thresholds for T3, the other nutrient thresholds (e.g., fiber) for T3 were determined by matching single nutrient compliance levels, as described earlier. In the cases where the WHO NPs were not used as benchmark for T3, T3 was determined following the same procedure as described subsequently for T2 and T4. To determine T2, C2 was first defined by the average of C1 and C3. In case T3 had not yet been defined, C2 was determined by dividing the non-compliant products into four equal portions, or C2 = C1 + (100% − C1)/4. Then the T2 nutrient thresholds were determined such that the combined nutrient compliance level would be C2, and the nutrient compliance levels for individual nutrients would not differ by more than 5% from each other. In a similar way, T4 and, if not defined already with input from WHO, also T3, were determined.

### 2.2. Criteria Validation

Nutrient profiling systems need validation to check their ability to correctly identify healthier food products, which is also recommended by the WHO [19]. This is essential to ensure a scientific basis and for consumers’ trust in the system. However, validation is one of the most challenging aspects of the NP model’s development [20,21,22]. Since no gold standard for identifying healthy food products exists, a possible option is an indirect validation of the NPS. Such a validation consists of testing the criteria’s ability to classify foods considered healthier or less healthy by nutrition professionals, established health standards such as the national dietary guidelines, or specific diet quality measures (e.g., the Healthy Eating Index) [20,21,22]. This method is also called convergent validity [13]. For the validation of the thresholds, lists of healthier and less healthy indicator foods for basic food groups, and less healthy food for non-basic food groups were compiled. These indicator foods are foods recommended or discouraged by food-based dietary guidelines from the three world regions (Europe, Africa, and Asia) that Choices focuses on. In particular, the dietary guidelines of UK, Germany, Spain, The Netherlands (Europe), Nigeria, South-Africa, Kenya (Africa), and India, Indonesia, and The Philippines (Asia) have been used. No indicator foods for the food groups water, fresh fruit, and vegetables, and unprocessed seafoods were collected, since all products in these groups are automatically considered as healthier. For each region, each indicator food has been classified as healthier or less healthy, supervised by one of the authors from the respective region.

Using the nutrition composition of each indicator food, each of them was classified into one of the five levels, based on the Choices intermediate thresholds. When healthier foods were classified into levels 1 or 2, these products were considered as a match. Similarly, less healthy foods were expected to be classified as levels 4 or 5 for basic foods, and as levels 3, 4, and 5 for non-basic foods. All other judgement classification combinations were considered a mismatch and investigated further, to identify the nutrient(s) causing the mismatch and to understand whether there was a need to adapt the intermediate thresholds for a specific food group. For an overview of the almost 300 indicator foods used for Africa, Asia, and Europe, see Appendix A.

### 2.3. Targeted Consultation

As further validation of the criteria, we contacted a series of stakeholders potentially interested or involved in applying Choices’ thresholds, to collect their views and feedback. These stakeholders are scientists and other food and nutrition experts working for the UN, governments, NGOs, academia, or the food industry. This process is commonly known as a ‘targeted consultation’ and it started with preparing a list of 54 stakeholders from the different sectors mentioned above and from the geographical areas where Choices operates (i.e., Europe, Africa, and South-East Asia). These stakeholders received an invitation email, together with a link to a questionnaire (included as Appendix A), a table depicting the criteria after the validation process, and an explanatory video summarizing the methodology used to develop the criteria. More specifically, stakeholders were requested to express their opinion on the food grouping and the nutrients used for establishing the criteria, the database used to calculate the criteria, and the thresholds identified.

## 3. Results

The 23 basic and 10 non-basic food groups Choices used for defining the present criteria are unchanged compared to the previous categorization used for the Choices logo criteria [8].

### 3.1. Intermediate Five-Level Criteria

The procedure depicted in Figure 3 was followed, to determine the intermediate thresholds T1–T4 for each of the food groups. These intermediate results are listed in Appendix A for basic food groups and in Appendix A for non-basic food groups.

For basic food groups, the T1 thresholds were defined in line with Choices’ previous criteria, with few exceptions. More restrictive thresholds were set for six food groups: total sugar for processed vegetables, processed fruit, milk products, sandwiches and rolls; sodium for oils, fats, and spreads; and SAFA for oils, fats, and spreads, as well as sandwiches and rolls. These decisions were made based on the relatively high overall compliance level C1 and based on compliance with foods that are generally perceived as healthy, such as beetroot, milk, and olive oil, as indicated in Appendix A. Input from the WHO NPSs to define T3 was used in 12 of the 23 basic food groups. In most cases the food group definitions of the WHO were considered sufficiently similar to Choices, with three exceptions. First, the WHO food group ‘fresh meat’ has a broader definition that includes fish and does not specify nutrient thresholds for sodium or SAFA, which are used by Choices. Second, the Choices food group ‘flavored noodles and pasta’ does not exist in the WHO NPSs as a separate food group, but is included in ‘ready meals’. Third, the Choices food group ‘insects’ is not specified by any of the WHO NPSs. Despite similar product group definitions, the WHO thresholds were not used for processed fruits, nuts and seeds, bread, breakfast cereals, and cheese (see also Appendix A). For processed fruits the WHO sugar level of 10/100 g was considered too restrictive, as this would imply a T1 sugar <7/100 g, exceeding natural sugar levels in many fruits. For nuts and seeds, the WHO benchmark for sodium levels of 50 mg/100 g is much lower than that considered insignificant (100 mg/100 g). For bread, the WHO sodium benchmark 0.48/100 g would imply a T1 sodium <0.32/100 g, which seemed too restrictive in comparison with the previous logo criteria (0.45/100 g), and was not used as part of the intermediate results. Similarly, for breakfast cereals, the WHO sugar benchmark of 15/100 g seemed too restrictive in comparison with the previous logo criteria (17/100 g) and was not used as part of the intermediate results. Similarly, for cheese, the WHO sodium benchmark of 0.6/100 g would imply a T1 sodium <0.4/100 g, which seemed too restrictive in comparison with the previous logo criteria and was therefore also not used as part of the intermediate results.

In the case of non-basic food products, T1 thresholds for the energy content of savory snacks equal to 500 kcal/100 g instead of 110 kcal/portion and for sweet snacks 220 kcal/100 g instead of 110 kcal/portion were defined, as standardized portion sizes are not available. The only non-basic food group for which T1 was set more strictly than the previous logo criteria is fruit and vegetable juices: a more restrictive sugar threshold T1 = 5/100 g than the logo threshold of 12/100 g was chosen to obtain an overall compliance C1 = 20%.

### 3.2. Validation by Indicator Foods

Appendix A list 65 indicator foods from Africa, 78 from Asia, and 147 from Europe. Furthermore, these tables include references to their product compositions; their classifications as healthier or less healthy products according to their national dietary guidelines; the classifications by the intermediate threshold levels; whether these classifications matched or not, and if not, what the explanation for the mismatch is; and whether this should lead to changes in the intermediate thresholds. For 60–70% of the products, there was a match between the judgements and the intermediate threshold levels. Many of the mismatches could be explained by higher or lower nutrient content levels than those expected by the authors. For example, cabbage sauerkraut was judged as healthier, but was classified at level 4 due to the sodium content of 0.6/100 g. Other causes for mismatches were differences between the dietary guidelines themselves or differences between the nutrition reference frame of the Choices criteria compared with several dietary guidelines. For example, in some dietary guidelines (ultra-) processed foods or canned foods are explicitly discouraged. Choices has not included any criterion related to processing.

### 3.3. Targeted Consultation

Choices conducted a targeted consultation with the previously defined stakeholders, collecting their feedback and suggestions, to finalize the criteria before publication. Appendix A gives a quantitative overview of the responses sent by the stakeholders.

Among the reasons for not agreeing with the choice of food groups, the two most common were because of the unclear classification of meat replacements, and the fact that non-dairy milk products were classified as non-basic food products despite their potential role in reducing the environmental impact of global diets. One stakeholder criticized the fact that both red and processed meat products were classified as basic food products, despite being classified as probably carcinogenic by the IARC [23].

Some stakeholders expressed the need for also considering more positive nutrients, whereas one pointed out that the choice of nutrients considered in each food group was not always clear, mentioning the example that there were criteria for sugar content for cheese, but no criteria for sodium in milk products. Some stakeholders also pointed out that the WHO criteria tended to be strict and should have been used for benchmarking T1 instead of T3.

Some concerns were expressed about the use of the George Institute database, which was judged not comprehensive enough by one stakeholder, who expressed concerns about a possible bias during the extrapolation process. One stakeholder pointed out that this database might not be relevant to all Asian countries or contain a similar amount of products for each country in this region. When asked about thresholds that they would change, some stakeholders mentioned that some criteria had ‘unequal jumps’ between one another (e.g., sodium in processed tubers from 0.100, 0.350, 0.400, and 1.600/100 g). One stakeholder pointed out that T1 should not be stricter than the logo criteria, whereas another wrote that T1 should not be stricter than the WHO criteria. Another one considered some thresholds were too lenient, citing the criteria for sodium in cheese and bread as an example. The criteria for milk products were considered too strict by one respondent, who also pointed out the same thing for the SAFA and sodium criteria for oils, spreads, and fats. One respondent pointed out that fiber levels were so strict that a regulated ‘source of’ claim would not be possible according to Choices criteria.

The remarks regarding food classification are useful and seem to make sense, but should be carefully considered, as for example not all non-dairy milk products and meat replacers contribute positive nutrients to diets, which is expected for products in basic food groups. To limit the scope of this work to the extension of the criteria, the Choices International Scientific Committee decided to stick with the Choices food group classifications and address the suggestions during the next criteria revision. The need to consider positive nutrients beyond fiber is fully justified, but is context- and geography-specific and could be included when customizing the international criteria to local or regional contexts.

### 3.4. Finalization Five-Level Criteria 

Based on the validation with indicator foods and the feedback from stakeholders, for some food groups the intermediate thresholds were adapted. For example, the intermediate thresholds did not sufficiently differentiate between whole grain, brown, and white cereal-based products. Appendix A describes the thresholds that were modified and the reasons for such changes. For three food groups, breakfast cereals, bread, and cheeses, it was decided to follow the WHO benchmarks for T3, despite the fact that this was first considered unrealistic when determining the intermediate thresholds. For breakfast cereals, validation showed that the sugar criteria were too lenient and it was decided to use the WHO sugar benchmark (15/100 g) as input for T3 and adapt T1 sugar to 10/100 g. This results in low compliance levels of C1 = 5% and C3 = 14%, emphasizing the need for product reformulation. For bread, the implication of using the WHO sodium benchmark of 0.48/100 g for T3, is that T1 = 0.32/100 g (50% stricter than T3), resulting in a very low compliance level of C1 = 2.5%. However, as bread is an important source of sodium in the diet, and the fact that it is technically possible to achieve the T1 threshold, such a low compliance level is considered justified, and stricter sodium thresholds provide an incentive for reformulation. For cheese, the lower sodium benchmarks imply that mainly soft cheeses would be able to qualify as healthier and that hard cheeses, which are traditionally prepared with higher salt levels, would generally not qualify as healthier. The product compliance C3 of products in the database against T3 (based on WHO sodium threshold 0.6/100 g) is only 1%. Nevertheless, it was decided to follow the WHO benchmark, despite the fact that with such stringent thresholds it may be difficult (but not impossible) to reformulate hard cheeses to be classified as healthier (level 1 or 2). On the other hand, it is also believed that reformulating hard cheeses, from e.g., level 4 to 3 is be beneficial, and that with the multi-level structure there are still sufficient incentives for producers to do so. In the absence of a good list of indicator foods for non-basic food groups, it was decided to leave the threshold levels for non-basic food groups unchanged. These changes resulted in the final Choices five-level criteria (4 threshold levels T1–T4) for basic and non-basic food groups that are listed in Table 1.

The product compliance levels C1–C4, i.e., the percentage of products in each food group (in total *N*) in the database that complied with all nutrient criteria for T1–T4, respectively, are provided in Appendix A. The distribution of products in each product group over the five levels L1–L5 is illustrated in Figure 5 for basic food groups and Figure 6 for non-basic food groups. It is evident that by using the logo criteria and WHO NPSs as a benchmark for T1 and T3, the distributions over the levels L1–L5 are no longer equal. Some food groups, such as fresh and processed fruit and vegetables, legumes, plain water, tea and coffee, and plain tubers, contain a relatively high percentage of healthier products than other food groups, such as flavored noodles and pasta, bread, cheese, or breakfast cereals. For non-basic foods, where the WHO benchmark was not available, the distribution is much more even.

## 4. Discussion

This paper describes the methodology used to extend the Choices criteria from a single threshold to four thresholds (T1 to T4), defining five levels of descending healthiness. This process, based on the existing Choices nutrient profiling methodology, considered the previous criteria and compliance levels, using a database of over 64,700 food products from basic food groups and non-basic food groups from eight countries worldwide developed by the George Institute in Australia [8], as well as various WHO NPS models for restriction of marketing to children. The five-level criteria were initially based on 20–40–60–80% compliance levels of products in the database, only. However, this approach lacked an alignment with Choices logo criteria and, more importantly, led to threshold levels that were not always coherent with indicator foods. Therefore, the Choices logo criteria were used as a benchmark for T1, and WHO NPSs were instead used for defining T3. The decision of whether these should be used for T1 and T3 could not be generalized to all food groups, but was taken for each food group, based on the desirability of using these benchmarks, but also after a reality check with packaged and non-packaged food products available in markets.

The criteria for the basic food groups were validated by testing them with a list of ‘indicator foods’. These were foods recommended by dietary guidelines from regions where Choices has a presence. Changes to the thresholds that followed the validation were made when the indicator foods and the classification of these foods according to the five-level criteria revealed some distortions. Finally, we collected feedback from stakeholders through a targeted consultation. The above process was supervised by an international standing committee of leading independent scientists, i.e., the Choices International Scientific Committee.

### 4.1. Nutrient Profiling vs. Dietary Guidelines

Food based dietary guidelines (FBDGs) represent a common tool to guide consumers’ dietary choices. However, we believe that NPSs can add value and coexist with FBDGs, with the aim of facilitating the adoption of healthier food choices by consumers. FBDGs were used as a guidance to define food groups for the Choices criteria [24]. A challenge in this work was to limit the number of food groups, while being sufficiently specific to distinguish food groups with distinct health properties. The main goal of official dietary guidelines is to summarize a large amount of evidence on diet and health and to use this knowledge to formulate culture-adapted actionable recommendations. In addition to influencing consumers’ dietary behavior, official dietary guidelines are also supposed to guide health and nutrition policies and strategies. FBDGs vary between countries. Not all dietary guidelines specify quantitative aspects that would facilitate their comparison and application in different contexts. In many guidelines, less attention is paid to foods, beverages, or nutrients to be limited (high-calorie foods, alcoholic beverages, salt) than to foods for which consumption is encouraged [25]. A main issue is also that processed foods are often not sufficiently addressed by FBDGs, or their consumption is only generically discouraged, without differentiating based on nutrient content. An important benefit of NPSs is that they also address processed food products. These are usually difficult to classify based on the grouping system adopted by FBDGs, either because they are a mix of different basic food products (e.g., pizza is made of grains, vegetable, and sometimes meat or fish) or because they contain added unhealthy ingredients (e.g., sugar or salt in canned fruits and vegetables). Additionally, not all food products within the same food group are equally healthy. NPSs offer the possibility of identifying, within the same food group, products with a more favorable composition, for instance a lower sodium or sugar content. NPSs discriminate healthier from less healthy mixed food products, including those that contain ingredients from different food groups (e.g., chicken noodle soup). Therefore, FBDGs and NPSs can complement each other and be used in different settings and for different purposes.

In many countries, authorities have increasingly recognized that nutrition policies require objective and reproducible methods to evaluate food products’ nutritional quality [19,26,27,28]. Consequently, the primary justification for developing nutrient profiling models is to provide rules for ranking food products according to their nutritional value [29] and to partially overcome the complexity of evaluating diets in their entirety. Another benefit of NPS is that the food industry can also use these models as guidance for food product reformulation [4]. Surprisingly, according to a recent review published by Labonté et al. [6], only 6 out of 78 of the reviewed NP models (including the Choices’ previous criteria) aimed at promoting food product reformulation.

However, to become effective public health tools, NPSs must also embrace the specificities of the context in which they are applied. More specifically, NPSs need to consider cultural differences, since healthier food products belonging to a balanced diet differ depending on the geographical region. For instance, meat provides nutrients that are usually lacking in the diet adopted by LMIC populations, whereas populations in high-income countries already consume excessive amounts [30]. Therefore, it is relevant to mention that the criteria described in this paper will be adapted to consider the traditions, nutritional issues, and cultural differences of the countries in which Choices is present.

Another reason why the Choices’ NPS can be considered an effective NP model is that it does not use the same thresholds for all food groups. Foods considered part of a healthy diet, such as nuts and seeds, usually score low when evaluated based on their saturated (or total fat) content, despite their favorable effects in terms of CVD prevention [31]. This Choices NP model developed specific thresholds for each food group, to consider the specific nutrition characteristics of each food group.

### 4.2. Strengths and Limitations

This work has both strengths and limitations. As for the limitations, we believe that one of the most important aspects of this work is that an official and direct validation (e.g., comparison with a gold standard) has never been defined for NPSs and, therefore, this process is usually performed using an indirect approach (such as testing the NPS against expert opinions or evaluating food products assumed to be healthy). Although this kind of validation can take different forms (content, face, predictive, or convergent validity), many existing NP models have never been validated, or their validation was not published [19,32,33,34]. Instead, the validation of the Choices five-level criteria made it possible to test their performance against marketed products in different regions and to adjust them considering local dietary guidelines and product particularities (e.g., SAFA content in milk products was relaxed to make full-fat milk compliant with level 3 and full-fat quark at level 4). The validation also helped test how the criteria performed in each food group, particularly those frequently used by some consumers because of cultural habits (e.g., dark sauces). Notably, the validation considerably reduced food product misclassification between indicator foods and the criteria, and, while validating the Choices criteria, it also became clear that some thresholds were too lenient (e.g., sugar content in breakfast cereals). These thresholds were modified to become more restrictive, and, as a result, the combined nutrient compliance levels C1–C3 are only 5, 9, and 14%, respectively; a clear sign that many products in the breakfast cereals food group need further reformulation to be considered as healthier.

The fact that it was not possible to test the effects of the application of these criteria on public health through a formal epidemiological study represents another limitation; hopefully to be addressed in the future. However, other NPSs developed based on similar principles to Choices, were also validated using data from epidemiological cohorts. As an example, consuming foods with higher Nutri-Score levels (i.e., of lower nutritional value) has recently been shown to be directly associated with a higher risk of mortality (overall and cancer mortality) [35]. In addition, the Food Standards Agency (FSA) NPS was found to be directly associated with the risk of metabolic syndrome in a cohort of middle-aged individuals [36], with a higher weight and BMI gain [37], and with an increased cardiovascular risk [38].

Choices validation using indicator foods also revealed that the perception of healthiness by scientists often deviates from the reality of the nutrient composition of food products. It is also realized that further validation using indicator foods is necessary. For example, concentrated tinned tomato puree, used as a healthy indicator food for processed vegetables, contains a sugar content of 13/100 g, and as a result is classified at level 5. Further validation studies are necessary to determine whether better threshold values are feasible. Furthermore, due to the difficulty of finding indicator foods for non-basic food groups in FBDGs, the validation was limited to basic-food groups.

The use of a large database of food products from middle- and high-income countries, worldwide, containing hundreds, to tens of thousands of food products per food group, including nutrition composition data that has been checked by the scientists of the George Institute in Australia, can be considered both as a strength and as a limitation. On the one hand, the development of a good NPS needs a representative food composition database, and the George Institute database is amongst the best available. However, it includes food products from only eight countries and is limited to packaged foods. Another important limitation is the fact that the George Institute database, used to calculate compliance levels, is incomplete in relation to both food products and nutrients, the information it contains are not always up to date, and it does not take differences in product sales and consumption into account.

In addition, the response rate to the targeted consultation was rather low, since less than one third of those invited agreed to participate. However, those who participated contributed with many comments that will be considered during future developments of the Choices criteria.

As for the strengths of this work, the definition of five healthiness levels provides a clear system by which to classify food products in different levels of healthiness and facilitates the comparison with other NPSs (e.g., Nutri-Score and Health Star Rating). A five-level system is flexible and can be easily rearranged into two or three levels for specific policy purposes, as shown in Figure 2. Among the strengths of this work, it should be mentioned that the full criteria setting process took place under the supervision and leadership of the independent Choices International Scientific Committee. This committee includes leading food and nutrition scientists from four continents, who have an independent position, in order to avoid any conflict of interest from the food industry. In addition, feedback was obtained from stakeholders from different geographical areas and sectors, including academia, NGOs, government agencies, and the food industry. That the criteria are food group-specific and threshold based represents another important strength of Choices’ work, as the effects of each nutrient can be modulated by the specific food composition [39]. The Choices NPS is adaptable to national circumstances and can be expanded to include more positive nutrients, which makes it the only NPS with an explicitly global ambition. Finally, the Choices criteria are revised every four years, to keep pace with developments in nutrition science and to stimulate food product reformulation.

## 5. Conclusions and Future Perspectives

The present study allowed the definition a new international NPS that can be applied to different contexts and support a variety of health policies, to prevent both undernutrition and obesity (i.e., double duty actions). This new NPS has the necessary flexibility to be adapted and applied worldwide. Choices realizes that its position has evolved significantly, from its earlier focus on best-in-class products and positive logos alone, to an NPS supporting multiple food system actions. However, this evolution of the criteria and its recommended use are all based on Choices’ strong science base, global scope, pragmatic approach, and fruitful collaboration with partners worldwide. More work will be invested to regularly evaluate, develop, and validate these criteria further. Choices also aims to establish epidemiologic collaborations, to test the long-term effects and the potential impact on environmental sustainability of diets resulting from the application of the Choices criteria by consumers. The targeted consultation revealed the necessity to reconsider the classification of animal product replacements, such as milk and meat replacements, which represent foods that could potentially improve the environmental sustainability of food systems. Finally, Choices aims to continue its work of developing thresholds for positive nutrients besides fiber, since these nutrients are fundamental for the prevention of malnutrition in many countries.

## Figures and Tables

**Figure 1 nutrients-13-04509-f001:**
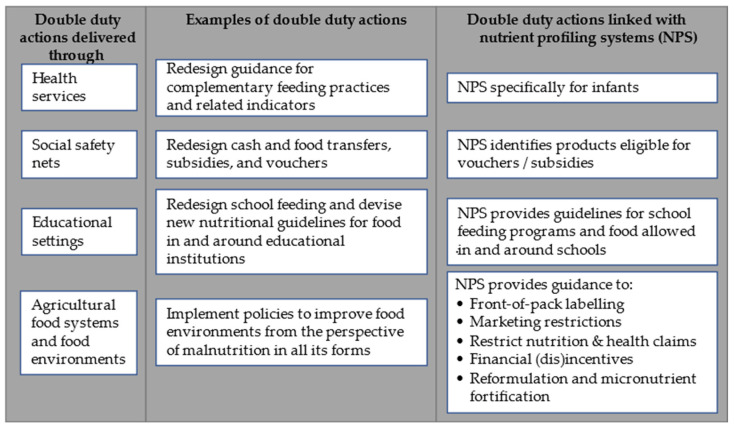
Examples of double-duty actions, addressing undernutrition, obesity, and diet-related non-communicable diseases, listed by Hawkes et al. [5], and how nutrient profiling systems (NPSs) could support them (authors’ analysis).

**Figure 2 nutrients-13-04509-f002:**
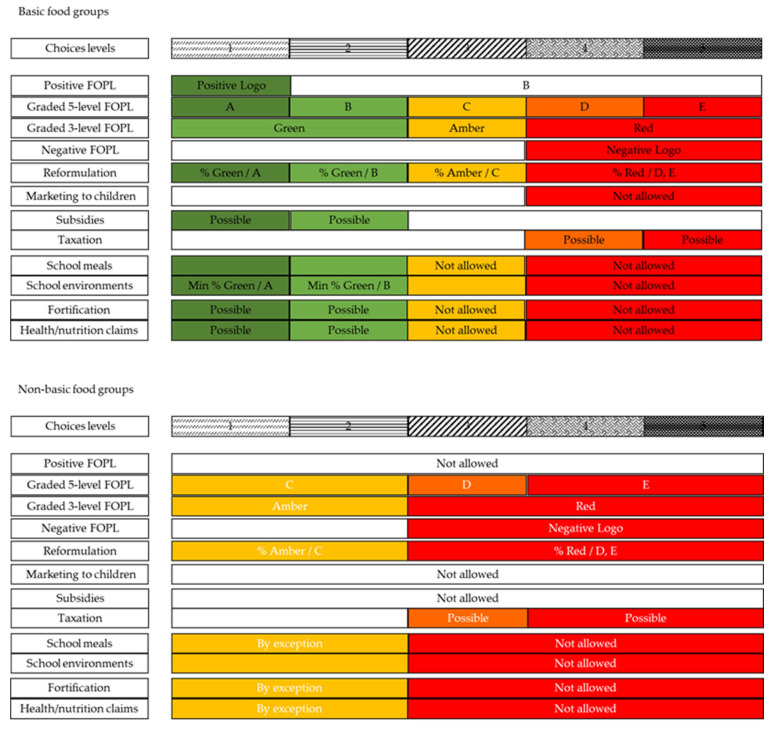
Recommended application of Choices five-level criteria for basic and non-basic food groups, to support multiple food system actions. FOPL = front-of pack label.

**Figure 3 nutrients-13-04509-f003:**
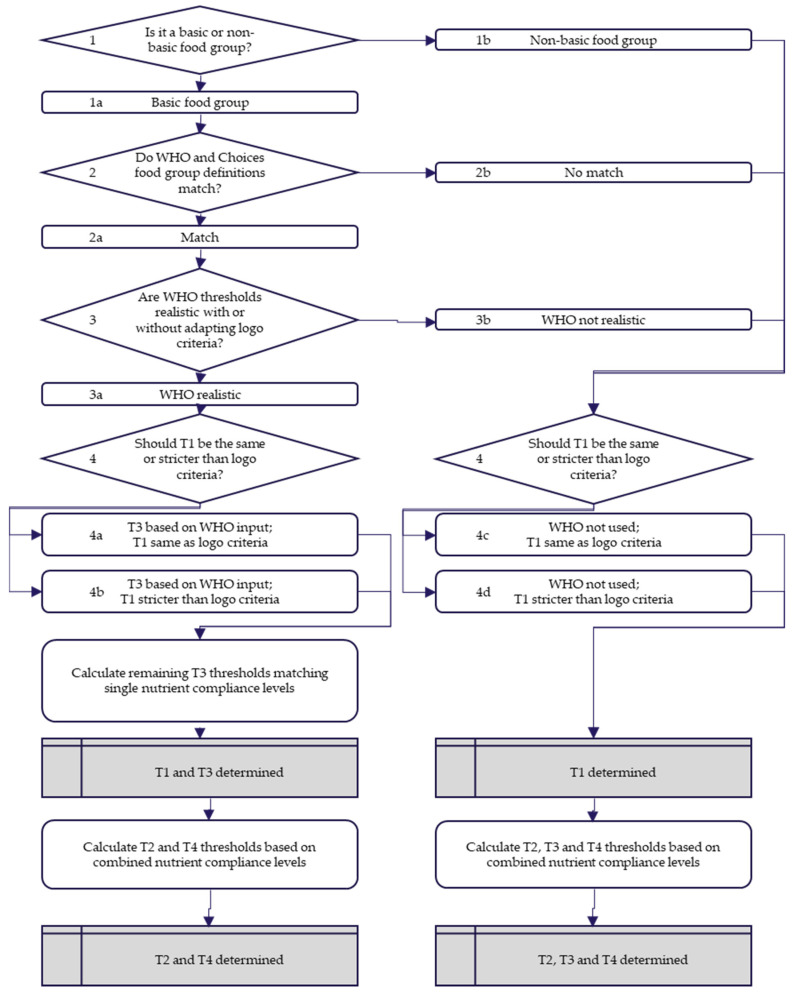
Decision tree describing the process of determining each threshold, T1–T4. The decision of whether the WHO thresholds were considered realistic is taken by assessing compliance levels C1 and C3 of products in the database. This determines at least T1 and, if WHO is used as input for sodium and/or sugar, also these nutrient thresholds for T3. The remaining nutrient thresholds for T3 are then determined by matching single nutrient compliance levels. Finally, T2 and T4, and if WHO was not used, also T3, are determined by interpolation of the combined compliance levels.

**Figure 4 nutrients-13-04509-f004:**
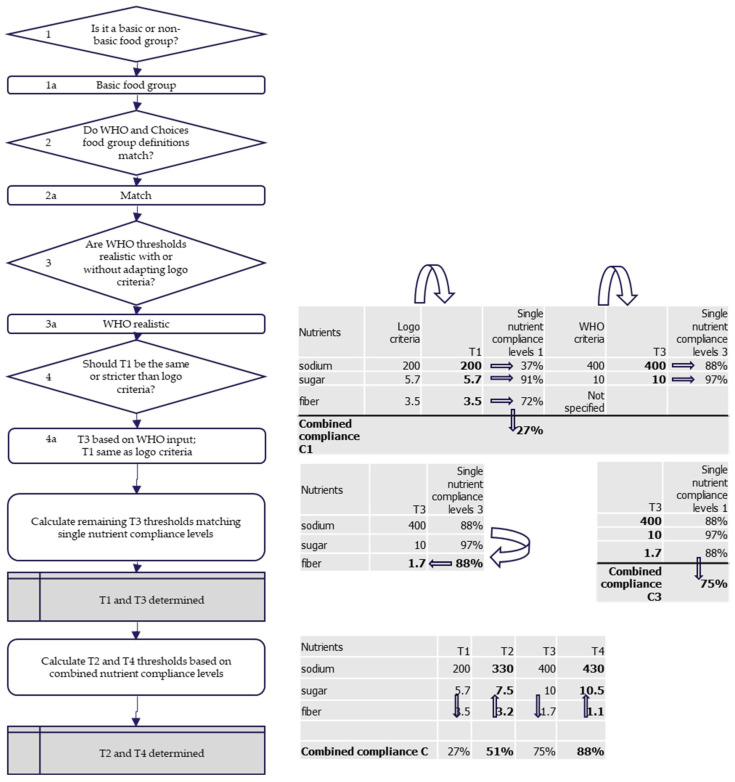
Illustration of how the thresholds T1–T4 for sodium, sugar, and fiber were determined for the food group ‘processed beans and legumes’. This food group is a basic food group (step 1) and the WHO uses a similar food group definition (step 2). T1 sodium and sugar thresholds are at least 50% stricter than WHO criteria, and a combined compliance level C1 = 27%, i.e., 27% of products in the database complying with these T1 thresholds, was considered realistic (step 3 and 4). Therefore, it was decided to use the logo criteria for T1 and use WHO as input for T3 (decision 4a). As the WHO does not provide a threshold for fiber, T3 for fiber is determined by matching single nutrient compliance (88%) with other single nutrient compliance levels, resulting in a value for T3 fiber = 1.7. Now all T3 threshold values have been determined, the combined compliance C3 = 75% is calculated. C1 and C3 are then used to determine C2 (51%) and C4 (88%), as described in the text, which are used to determine T2 and T4.

**Figure 5 nutrients-13-04509-f005:**
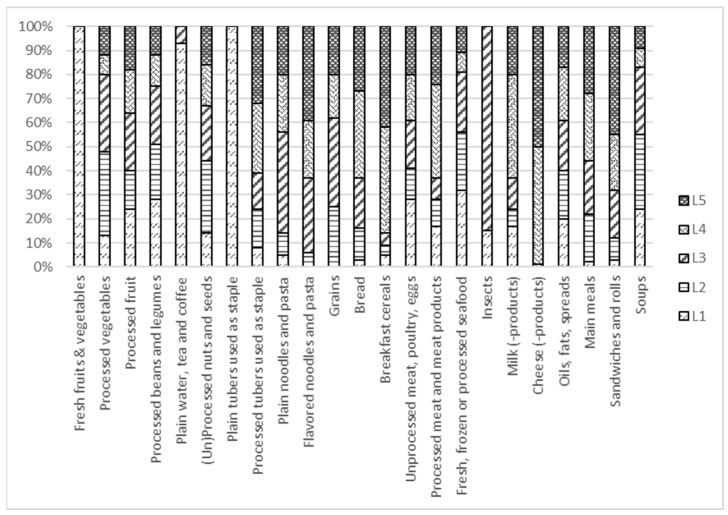
Distribution of products in the database over levels L1–L5 for basic food groups. Due to the use of WHO benchmarks and the validation against indicator foods, a relatively high percentage of fresh and processed fruit and vegetables, legumes, plain water, tea and coffee, and plain tubers are classified as healthier, whereas other food groups, such as flavored noodles and pasta, breakfast cereals, bread, and cheese contain more less healthy products.

**Figure 6 nutrients-13-04509-f006:**
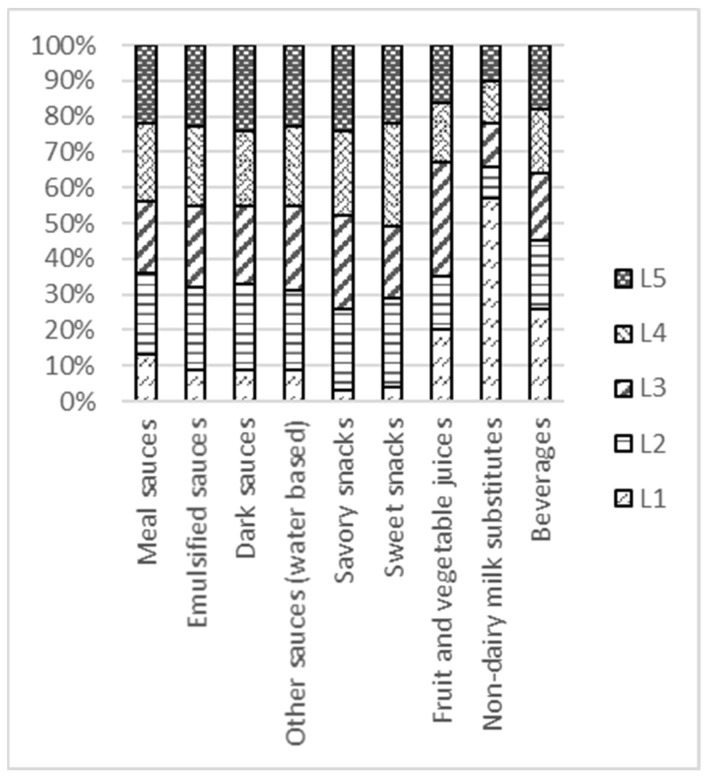
Distribution of products in the database over levels L1–L5 for non-basic food groups. As WHO NPs did not provide a benchmark for T3, the distribution over levels 1–5 is much more even than for basic food groups.

**Table 1 nutrients-13-04509-t001:** Choices five-level criteria for basic and non-basic food groups after the validation and consultation process. To comply with a certain threshold, product nutrient content should be ≤ (for fiber ≥) than listed levels for all nutrients. When nutrient levels are not provided, they are considered non-critical for that food group and insignificant. T_1_–T_4_ = Thresholds 1–4, SAFA = saturated fatty acids, iTFA = industrially-produced trans fatty acids.

	Food Group	T_1_						T_2_						T_3_						T_4_					
		SAFA	iTFA	Sodium	Sugar	Fiber	Energy	SAFA	iTFA	Sodium	Sugar	Fiber	Energy	SAFA	iTFA	Sodium	Sugar	Fiber	Energy	SAFA	iTFA	Sodium	Sugar	Fiber	Energy
		g/100 g					kcal/100 g	g/100 g					kcal/100 g	g/100 g					kcal/100 g	g/100 g					kcal/100 g
**Basic food groups**																									
Fruits and vegetables	Fresh fruits and vegetables	All compliant																							
	Processed vegetables			0.10	7.0	1				0.25	8.5	0.9				0.40	10.0	0.8				0.65	11.0	0.7	
	Processed fruit	1.1			11.5	1		2			12.5	0.9		3			14.0	0.8		4			19.0	0.7	
	Processed beans and legumes			0.20	5.7	3.5				0.33	7.5	3.2				0.40	10.0	1.7				0.43	10.5	1.1	
Water	Plain water, tea and coffee			0.02						0.02															
Nuts and seeds	(Un)Processed nuts and seeds	10.0		0.10	7.5			16.0		0.43	14.0			18.0		0.55	30.0			20.0		0.73	36.0		
Sources of complex carbohydrates	Plain tubers used as staple	All compliant																							
	Processed tubers used as staple	1.1		0.10	3.0	2.7		3.0		0.35	6.5	2.2		4.0		0.40	10.0	1.5		8.0		1.60	12.0	0.8	
	Plain noodles and pasta			0.10	4.0	6.0				0.20	4.2	2.8				0.48	5.0	1.0				0.80	6.0	0.5	
	Flavored noodles and pasta	2.0		0.50	4.0	6.0		3.5		0.93	4.2	2.8		6.5		1.20	5.0	1.0		8.0		1.50	6.0	0.5	
	Grains	1.2		0.10	4.5	6.0		1.5		0.23	6.0	2.8		1.8		0.48	10.0	1.0		4.0		1.40	12.0	0.5	
	Bread	1.1		0.32	6.0	6.0		1.8		0.40	6.5	2.8		3.5		0.48	9.0	1.0		6.0		0.85	15.0	0.5	
	Breakfast cereals	3.0		0.40	10.0	6.0		3.2		0.50	14.0	2.8		3.3		0.64	15.0	1.0		4.2		0.68	26.0	0.5	
Meat, fish, poultry, and eggs	Unprocessed meat, poultry, eggs	3.2		0.15				3.7		0.17				5.3		0.40				7.5		0.68			
	Processed meat and meat products	5.0		0.45				6.0		0.60				8.0		0.68				10.0		1.30			
	Fresh, frozen or processed seafood	6.0		0.30				6.5		0.43				7.0		0.68				7.5		1.10			
	Insects	3.2		0.20				3.2		0.20															
Dairy	Milk (-products)	1.4			6.0			1.7			8.0			2.7			10.0			6.0			14.0		
	Cheese (-products)	7.5		0.40				8.5		0.50				10.0		0.60				19.0		1.20	6.0		
Oils, fats and fat containing spreads	Oils, fats, spreads	16.0	0.5	0.10				30.0	0.5	0.35				36.0	0.5	0.52				55.0	0.5	0.75			
Meals	Main meals	2.0		0.24	5.0	2.4	190 kcal/100 g and 600 kcal/portion	3.0		0.34	7.0	1.4	200 kcal/100 g and 600 kcal/portion	4.0		0.40	10.0	1.0	225	5.0		0.53	11.0	0.8	275
	Sandwiches and rolls	2.0		0.45	5.0	2.4	190 kcal/100 g and 350 kcal/portion	3.0		0.57	7.0	1.4	215 kcal/100 g and 350 kcal/portion	4.0		0.62	10.0	1.0	225	5.0		0.80	11.0	0.8	275
	Soups	1.1		0.25	4.0			2.0		0.29	5.0			3.5		0.35	9.0			4.0		0.39	10.0		
**Non-basic food groups**																									
Sauces	Meal sauces	1.1		0.40	6			1.3		0.70	8			2.5		2.20	16			6.0		4.50	26		
	Emulsified sauces	3		0.70	10		350	4.5		1.00	12		380	6		1.20	17		550	8		1.80	21		650
	Dark sauces			3.00	16					5.50	20					6.50	25.5					7.75	35		
	Other sauces (water based)			0.75	16.0		100			0.80	25.0		130			0.90	31.0		150			1.08	39.0		190
Snacks	Savory snacks	4.0	0.4	0.40	4.0		500	7.0	0.5	0.79	6.5		535	9.0	0.5	0.88	9.0		540	13.0	0.5	1.00	16.0		570
	Sweet snacks	6.0		0.20	20.0		220	12.0		0.22	45.0		475	16.5		0.31	55.0		510	20.0	0.4	0.41	62.0		550
Liquids	Fruit and vegetable juices				5.0						8.0						10.0						11.0		
	Non-dairy milk substitutes	1.1		0.10	5.0			1.5		0.11	6.0			2		0.12	7.2			5.5		0.13	9.0		
	Beverages				2.5						5.5						8.0						11.5		
Other	All other products	1.1 or 10 en%	0.1 or 1 en%	0.10	2.5 or 10 en%			1.1 or 10 en%	0.1 or 1 en%	0.10	2.5 or 10 en%														

## Data Availability

All additional data are included in the Appendix A.

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
