# Peer review of "Development of the Choices 5-Level Criteria to Support Multiple Food System Actions"

_nutrients, 2021, doi:10.3390/nu13124509_

Round 1
Reviewer 1 Report
I read the article "Development of the Choices 5-Level Criteria to Support Multiple Food System Actions" with interest. I have just a few comments:
The objectives of this work must be clearly stated in the abstract and in a separate section of the text.
Line 69-70 multiple interventions to promote and discourage the consumption of healthier and less healthy foods. - I would suggest writing "multiple interventions to promote the consumption of healthier and discourage the consumption of less healthy foods".
line 80 "The Choices NPS was developed to identify food products eligible for a positive FOPL, facilitating consumers to select healthier food options and at encouraging..." - to encourage?
I find figure 2 confusing, especially the part where 3b "crosses over" to the other side. Does 3b means step 3 for non-basic food groups? If so, there should be a line connecting it with the previous steps for this kind of product.
Choices conducted a targeted consultation with earlier defined stakeholders collecting their feedback and suggestions to finalize the criteria before publications. - "publication"?
Tables: please write any abbreviations in full in the legend
I would advise not to separate the discussion in section, and to use sentences to introduce the paragraph instead. The conclusions may be in a separate section.
Reviewer 2 Report
The subject of the work is interesting, current and necessary in today's society. It seems to me that it could be a good work to publish in the journal Nutrients. However, the description of the work lacks a research approach, where a specific objective of the study is stated, the methodology followed to achieve this objective, and the results and conclusions obtained.
From my point of view, while the reader is clear about the context of the work, it is not easy to delimit the research objective.
My recommendations are along these lines:
- The abstract would be improved if it were rewritten to present the following sequence of information: context of the work, research objective, methodology, results and conclusions.
- Although the paper has current citations, few of them belong to research papers published in scientific journals. I suggest the authors to include citations from scientific journals with papers validating other Nutrients Profile Models.
- The objective of the work is not described in the introduction. This should be specified to improve the comprehension of the text.
- In the Targeted Consultation in section 2, the Delphi method is applied. The authors should name and describe this methodology correctly, providing bibliographic citations.
- It is in the discussion section that the objective of the work is understood more concretely. As I indicated in point 3, this should be specified more clearly in the introduction.
- The section "Conclusion and Future Perspectives" should include a clear description of the conclusions reached on the basis of the objective set out in the introduction.
- The paper has no a description of the limitations but it has them. I have not commented on the validation methodology, because I understand that the study has already been done and it does not make sense for me to propose any modification in this respect. However, I consider that the validation would have been more enriching if consumers had been included. In my view, the existing NPMs are not properly consumer-oriented and one way to find out whether they are understandable by consumers is to test them with them, considering their consumption behaviour and the context of purchase.
Round 2
Reviewer 2 Report
Accept article to be published